# Alcohol inhibits the metabolism of dimethyl fumarate to the active metabolite responsible for decreasing relapse frequency in the treatment of multiple sclerosis

Bing Yang[1], Robert B. Parker[1], Bernd Meibohm[2‡], Zaid H. Temrikar[2‡], Ashish Srivastava[2‡], S. Casey Laizure[1]*

1 Department of Clinical Pharmacy and Translational Science, College of Pharmacy, University of Tennessee Health Science Center, Memphis, Tennessee, United States of America, 2 Department of Pharmaceutical Sciences, College of Pharmacy, University of Tennessee Health Science Center, Memphis, Tennessee, United States of America

☯ These authors contributed equally to this work.
‡ BM, ZHT and AS also contributed equally to this work.
* claizure@uthsc.edu

**Data Availability Statement:** All relevant data are within the article and its Supporting Information files.

## Abstract

Dimethyl fumarate (DMF) is a first-line prodrug for the treatment of relapsing-remitting multiple sclerosis (RRMS) that is completely metabolized to monomethyl fumarate (MMF), the active metabolite, before reaching the systemic circulation. Its metabolism has been proposed to be due to ubiquitous esterases in the intestines and other tissues, but the specific enzymes involved are unknown. We hypothesized based on its structure and extensive presystemic metabolism that DMF would be a carboxylesterase substrate subject to interaction with alcohol. We sought to determine the enzymes(s) responsible for the extensive presystemic metabolism of DMF to MMF and the effect of alcohol on its disposition by conducting metabolic incubation studies in human recombinant carboxylesterase-1 (CES1), carboxylesterase-2 (CES2) and human intestinal microsomes (HIM), and by performing a follow-up study in an in vivo mouse model. The in vitro incubation studies demonstrated that DMF was only metabolized to MMF by CES1. Consistent with the incubation studies, the mouse pharmacokinetic study demonstrated that alcohol decreased the maximum concentration and area-under-the-curve of MMF in the plasma and the brain after dosing with DMF. We conclude that alcohol may markedly decrease exposure to the active MMF metabolite in the plasma and brain potentially decreasing the effectiveness of DMF in the treatment of RRMS.

## Introduction

Dimethyl fumarate (DMF) is first-line, oral therapy for the treatment of relapsing-remitting multiple sclerosis (RRMS). Clinical studies leading to the approval of DMF for the treatment

**Funding:** These studies were funded by the National Institute of Neurological Disorders and Stroke grant R03NS116229 (PI: S. Casey Laizure) and NIH S10 grant 1S10OD016226-01A1 (PI: B. Meibohm). The mice used in this study were kindly provided by Dr. Phillip Potter, St. Jude Children's Research Hospital, Memphis, TN. The funders had no role in study design, data collection and analysis, decision to publish, or preparation of the manuscript.

**Competing interests:** The authors have declared that no competing interests exist.

of RRMS demonstrated that twice daily dosing resulted in about a 50% reduction in the annualized relapse rate [1]. After an oral dose, DMF is completely metabolized by hydrolysis to the purported active metabolite, monomethyl fumarate (MMF) before reaching the systemic circulation. In human subjects taking a typical oral dose of DMF(140 mg or 240 mg), only MMF can be quantified in the plasma with no detectable levels of DMF [2]. What specific enzyme(s) are responsible for this extensive presystemic hydrolysis of DMF to MMF is unknown with conflicting reports on the role of intestinal [3–5], plasma [1], and hepatic esterases [2]. Esterases with the capacity to completely hydrolyze DMF prior to reaching the systemic circulation would most likely be intestinal or hepatic esterases, which implicates carboxylesterase-1 (CES1), which is the most abundant esterase in the liver and carboxylesterase-2 (CES2) found primarily in the intestines [6]. Based on the chemical structure of DMF and the specificity of CES1 for esters with small alcohol groups [7] we hypothesized that DMF would be a substrate of the CES1 hepatic esterase and interact with alcohol. This hypothesis is supported by previous studies with other CES1-substrate drugs such as cocaine, oseltamivir, and methylphenidate demonstrating that alcohol inhibits CES1-mediated hydrolysis and results in the formation of novel alcohol-transesterified metabolites.

Alcohol inhibits CES1-mediated hydrolysis and results in the formation of novel metabolites through transesterification [8, 9]. Transesterification occurs when the ethyl group of alcohol displaces the ester alcohol group of the CES1 substrate instead of the ester alcohol group being displaced by the hydroxy group from water (hydrolysis). This results in the formation of a novel ethyl ester metabolite instead of the carboxylic acid formed by hydrolysis (Fig 1). Numerous CES1-substrate drugs including oseltamivir [10, 11], cocaine [12, 13], clopidogrel [8, 9], and methylphenidate [14, 15] have been shown to interact with alcohol resulting in the slower formation of the hydrolysis product and the formation of ethyl ester metabolites. This interaction between alcohol and CES1-substrate drugs has been a consistent finding demonstrated in studies conducted in vitro, in animal models, and in human studies [16].

We conducted a series of in vitro experiments and a study in a mouse model to characterize the metabolism of DMF by carboxylesterases and its interaction with alcohol to identify the primary metabolic pathway for DMF's hydrolysis to its active metabolite and the potential for alcohol consumption to affect its disposition.

## Methods

### Materials

DMF, di-ethyl fumarate (DEF), and alcohol (100%) were purchased from Sigma-Aldrich (St. Louis, MO, USA). MMF was purchased from Santa Cruz Biotechnology Inc (ChemCruz, Dallas, TX, USA). Tiopronin and oseltamivir-d3 were purchased from Toronto Research Chemicals Inc. (Toronto, Canada). Sodium bis (4-nitrophenyl) phosphate (BNPP) was purchased from Tokyo Chemical Industry Co. Ltd. (TCI, USA). Ethyl methyl fumarate (EMF) was synthesized by our Medicinal Chemistry Core laboratory at the University of Tennessee Health Science Center. HPLC-grade acetonitrile (ACN), methanol (MeOH), and dimethyl sulfoxide (DMSO) were purchased from Fisher Scientific (Waltham, MA, USA). Human recombinant CES1 (lot#7059001), CES2 (lot#7130003), and human intestinal microsomes (HIM) (lot#6188002) were purchased from Corning (Woburn, MA) and stored at -80°C until used. The mice used in this study were obtained from St. Jude Children's Research Hospital (Memphis, TN, USA). They are a hybrid offspring from C57BL/6J and DBA/2J mice having carboxylesterase-deficient plasma (Es1$^e$), which provides a rodent model of carboxylesterase drug metabolism that more closely mimics human carboxylesterase drug metabolism than typical strains of rodents [17].

**Fig 1. DMF hydrolysis and transesterification catalyzed by CES1.** CES1 catalyzed hydrolysis produces the active carboxylic acid metabolite, MMF, while CES1 catalyzed transesterification with alcohol ($CH_3CH_2$-OH) produces an ethyl ester, ethyl methyl fumarate.

## In vitro studies in CES1, CES2, and in HIM

The initial DMF concentration for incubation studies was 50 μM with a total final volume of 100 μL. All incubations were conducted at pH of 7.4 in a potassium phosphate buffer at 37°C. The reactions were terminated by adding 100 μL of cold ACN. The incubation times were 0, 5, 10, 20, and 30 minutes. All incubation studies were conducted in triplicate, except for the initial assessment of the hydrolysis of DMF in CES1, CES2, and HIM (Fig 2) in which each data point represents a single incubation.

## In vivo study in Es1$^e$ mice

The mice used in this study were a plasma carboxylesterase-deficient mouse strain (Es1$^e$) obtained from St. Jude Children's Research Hospital (Memphis, TN). This strain of mouse is a better model of human carboxylesterase drug metabolism because they have low plasma carboxylesterase activity which more closely mimics human CES1 drug metabolism [17]. The animal studies were approved by the University of Tennessee Health Science Center Institutional Animal Care and Use Committee and experiments were performed according to the Guide for the Care and Use of Animals. Es1$^e$ mice were housed in standard conditions (12 h light/dark cycle) and had ad libitum access to food and water prior to the study. The night before the study the mice were fasted (no more than 10 hours before dosing). A total of 60 mice (30 male and 30 female) (ages 12–18 weeks) were divided randomly into two groups: a group that was dosed with water followed 10 minutes later by a 100 mg/kg dose of DMF (Control group), and a group that was dosed with 3g/kg alcohol followed 10 minutes later by 100 mg/kg of DMF (Alcohol group). All doses were administered by oral gavage. The DMF vehicle was DMSO and PEG300 in a 19:1 v/v ratio. The dose of DMF was at the high end of the range used in previous mouse studies [18, 19] to ensure the MMF plasma concentrations could be quantified over a reasonable time frame, and the alcohol dose of 3 g/kg is the same dose as used in a previous mouse study of alcohol's interaction with methylphenidate, a CES1-substrate drug [20].

At pre-determined time points (5, 10, 20, 30, 45, 60, 90, 120, and 180 min after DMF administration) mice were heavily anesthetized with isoflurane to collect a single retro-orbital blood

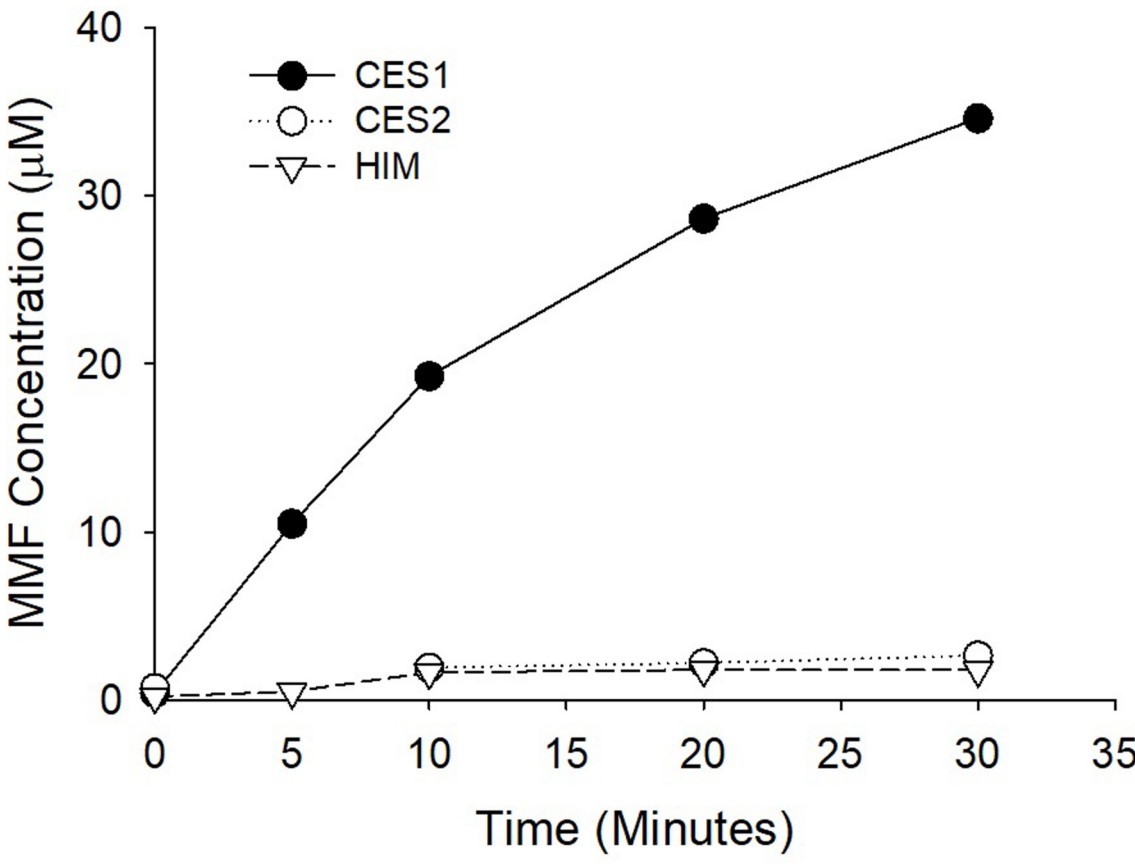

**Fig 2. The formation of MMF in CES1, CES2, and HIM.** DMF was incubated for increasing time periods (0, 5, 10, 20, and 30 minutes) in human recombinant CES1 and CES2, and HIM. The concentration of MMF at the end of each incubation was determined by LC-MS/MS. Only CES1 hydrolyzed DMF to its active MMF metabolite.

sample (approximate blood volume collected was 200 μL). Following blood sample collection, the mice were euthanized, and brain dissected from each mouse. Blood was immediately mixed with 2 μL of BNPP solution (250 mM in DMSO) to prevent hydrolysis [21] and centrifuged at 6000×g for 5 minutes at 4°C using plasma separation tubes (BD Microtainer). Whole-body perfusion was performed to drain the mouse blood from the systemic circulation and remove residual blood from the brain according to a previously described method [22]. The brain was dissected and washed in 0.9% saline and gently dried with tissue paper. The brain tissue was immediately homogenized in four volumes (average organ weight×4) of phosphate buffer saline pH 7.4 containing the stabilizing agent BNPP. All plasma and tissue samples were stored at -80°C until analysis.

### DMF and metabolite (MMF/EMF/DEF) assay by LC-MS/MS

DMF and its metabolites were quantified using a modification of a previously described assay [5]. Stock solutions of DMF, MMF, EMF, and DEF were prepared in DMSO at a concentration of 20 mg/mL. Stock solutions of the internal standard (IS), oseltamivir-d3 (OS-d3), were prepared in acetonitrile at a concentration of 1 mM. Stock solutions of the trapping reagent tiopronin for the four compounds to be detected (DMF/MMF/EMF/DEF) were prepared in 100 mM ammonium bicarbonate buffer solution (pH 9.0) at a concentration of 1 M. These stock solutions were stored at -80°C. All the preparation and processing procedures were completed

on top of wet ice and all organic solutions and buffer for dilution were stored at 4˚C before use. The working solutions of DMF, MMF, EMF, and DEF were prepared fresh in ammonium bicarbonate buffer solution (pH 9.0) for each calibration standard curve at 1, 5, 10, 50, 100, 500, 1000, 5000 ng/mL (DMF: 7, 35, 69, 347, 694, 3469, 6938,34690 nmole/L; MMF: 8, 38, 77, 384, 769, 3843, 7686, 38430 nmole/L), and for quality control samples (QCs) at 5, 100, 3000 ng/mL (DMF: 35, 694, 20814 nmole/L; MMF: 8, 77, 23058 nmole/L) in mice plasma and brain tissue. Each spiked mouse standard sample was 25 μL and consisted of 1 μL of 250 mM BNPP, 4 μL of a four-compound mixture of DMF, MMF, EMF, and DEF, and 20 μL of blank mouse plasma or brain tissue.

Spiked samples, and mouse plasma and tissue samples were thawed on wet ice and a 25 μL aliquot transferred to a 0.75 mL polypropylene tubes. The fresh working solution of 100 mM tiopronin was diluted with ammonium bicarbonate buffer solution (pH 9.0), and 50 μL (2 times the sample volume) was added to each sample tube. The sample was thoroughly vortexed for approximately 10 min at room temperature and allowed to remain on the top of wet ice for an additional 5 minutes. The working IS solution was prepared fresh in MeOH: ACN (50:50 v: v) with 0.1% formic acid, then 150 μL (6 times the sample volume) of it was added into each tube to precipitate proteins and provide a final IS concentration of 100 ng/ml. Then the sample tubes were vortex-mixed for approximately 30 seconds and centrifuged for 10 minutes (15000×g) at 4˚C. Finally, 50 μL of supernatant was transferred into a clean tube and added with 50 μL of 10 mM ammonium formate buffer (pH 3), and vortex-mixed for about 10 sec. Following centrifugation for 10 min at 15000×g and 4˚C, 1 μL of the sample was injected for analysis.

A SCIEX triple quadrupole 5500 mass spectrometer (Framingham, MA) interfaced with a Turbo Ion Spray / electrospray ionization (ESI) source with a Shimadzu Nexera LC-20AD XR liquid chromatograph, CTO-20AC column oven and SIL-20AC autosampler (Columbia, MD) was used in the positive ionization mode for the quantification of derivatized DMF and metabolites. An XSelect CSH C18 column (2.1 mm×75 mm, 3.5 μm particle size, Waters Co, Ireland) was used for separation. The column temperature was maintained at 40˚C and the sample compartment was kept at room temperature. Mobile phase A consisted of 5 mM ammonium formate with 0.05% formic acid (pH 3.5) and 1% methanol and mobile phase B consisted of methanol with 5 mM ammonium formate with 0.05% formic acid. The LC system was held at 10% B for 1 min followed by a linear gradient from 10% B to 100% B from 1.0 to 2.0 min. Then conditions were held at 100% B for 2.0 min to remove late eluting substances from the column, after which the system was returned to initial conditions at 4.10 min. The total run time including the sample load was approximately 4.5 min and the flow rate was maintained at a constant 0.5 mL/min throughout the run. A Diverter valve diverted the flow to the mass spectrometer from 1.0 to 3.9 min.

In derivatization with tiopronin [5], the MRM transitions of parent-son ion pairs for these compounds and their conditions of declustering potential (DP), collision energy (CE), and collision cell potential (CXP) are shown in Table 1. The mass spectrometric conditions were source temperature 600˚C, ion source voltage 5500 V, curtain gas (nitrogen) 20 psi (1.38 bar), nebulizing and drying gas 60 psi (4.14 bar), collision gas 8 psi (0.55 bar). Calibration plots of the peak area ratio of the analytes to the IS versus their corresponding concentrations were constructed and linear regression was performed to obtain a standard curve. The analytical detection methodology was validated for its sensitivity including lower limit of detection, accuracy, precision, stability, and carryover. The QC samples were determined in each run and used in estimating the bias and precision of the method. The data were acquired and analyzed using the proprietary software application Analyst® Version 1.6.3 and MultiQuant® Version 3.0.2 (Applied Biosystems/MDS-Sciex, Framingham, MA).

**Table 1. Multiple reaction monitoring.** Analyte-specific parameters for the LC-MS/MS detection of DMF, MMF, EME, DEF and the internal standard.

| Compound | Q1 (Da) | Q3 (Da) | DP (V) | CE (V) | CXP (V) |
|---|---|---|---|---|---|
| DMF | 308.15 | 216.00 | 141 | 21 | 12 |
| MMF | 294.15 | 216.00 | 60 | 19 | 12 |
| EMF | 322.20 | 216.10 | 101 | 21 | 12 |
| DEF | 336.20 | 216.00 | 66 | 23 | 12 |
| IS (OS-d3) | 316.20 | 211.10 | 21 | 19 | 18 |

Q1; parent ion; Q3: son ion; Da: Dalton; V: volt

## Pharmacokinetic analysis

The mean plasma concentration (n = 3) at each time point (5, 10, 20, 30, 45, 60, 90, 120, 180 minutes) was plotted. Using this plasma-concentration time profile the area-under-the-curve (AUC) and elimination rate constant (k) were estimated using noncompartmental analysis (WinNonlin® software 8.4 version, Phoenix, California, USA). The AUC was determined by summing the trapezoids from zero to the last plasma concentration and adding the area of the tail calculated by dividing the last plasma concentration by the estimated k. The estimate of k was determined by linear regression of the terminal slope.

## Statistical analysis

The change in MMF concentration produced by 200 mM of alcohol in incubations of DMF in recombinant CES1 was compared with incubations carried out in recombinant CES1 without alcohol by performing a two-tailed t-test on the MMF concentration at the end of the incubation. The differences in the plasma and brain concentrations of DMF and MMF between the Control and Alcohol groups in the mouse study were compared using a two-tailed t-test with a Benjamini-Hochberg Procedure to account for multiple comparisons [23].

## Results

### LC-MS/MS assay validation

The method allowed the simultaneous quantification of DMF, MMF and potential transesterification metabolites with retention times of 2.55 (MMF), 2.71 (DMF), 2.81 (EMF), 2.89 (DEF), and 2.94 (IS) minutes. All standard curves and quality control samples were made by spiking mouse plasma or tissue with known concentrations of DMF and MMF over a concentration range of 1 to 5,000 ng/ml (DMF 7–34690 nmole/L, MMF 8–38430 nmole/L). Standard curves were reproducible and linear ($r^2$ 0.998 for DMF and 0.997 for MMF) and intra- and inter-run (n = 3) were less than 20% at low concentrations and less than 15% at moderate and high concentrations.

### In vitro metabolism studies

DMF (50 μM) was incubated in CES1 and CES2, and in HIM. Only incubations in CES1 produced significant hydrolysis to MMF. At the longest incubation time of 30 minutes, incubations in CES1 resulted in a concentration of MMF that exceeded 30 μM while incubations in CES2 and HIM produced concentrations less than 3 μM of MMF at the maximum incubation time of 30 minutes (Fig 2). Fig 3 is a plot of increasing concentrations of alcohol versus MMF produced. Alcohol is a documented inhibitor of CES1 enzyme activity and resulted in a concentration-dependent inhibition of DMF hydrolysis to MMF with an $IC_{50}$ of 19.7 mM (0.091

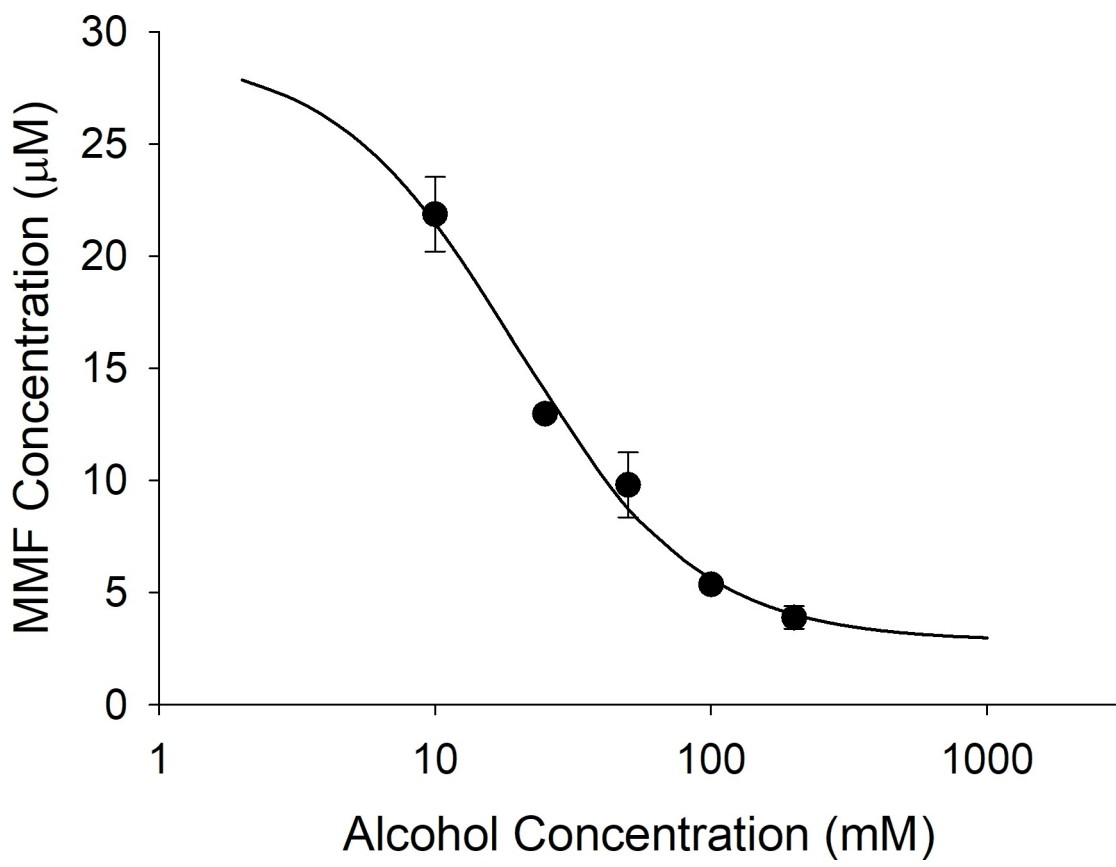

**Fig 3. The concentration-dependent inhibition of CES1 catalyzed hydrolysis by alcohol.** DMF (50 μM) was incubated for 30 minutes in recombinant human CES1 containing increasing concentrations of alcohol (10, 25, 50, 100, and 200 mM). The concentration of MMF at the end of the 30-minute incubation was determined by LC-MS/MS. The estimated $IC_{50}$ was 19.7 mM, which is equivalent to an alcohol concentration of 0.091 g/dL (legal limit for driving under the influence is 0.08 g/dL for point of reference).

g/dL). In the third in vitro study, a high concentration of alcohol (200 mM) was incubated for 60 minutes in CES1 with 50 μM of DMF to facilitate the formation of transesterification products that occur when the ethyl group from alcohol is transferred to the CES1 substrate instead of water resulting in an ethyl ester instead of a carboxylic acid metabolite. In the absence of alcohol in the CES1 incubation only MMF is formed and virtually all the initial DMF is accounted for at the end of the incubation by DMF and MMF. In the CES1 incubation with 200 mM alcohol, though DMF is similarly depleted after the 60-minute incubation period, only a relatively small proportion is converted to MMF, which does not account for the depletion of DMF (Fig 4).

### DMF alcohol study in Es1$^e$ mice

This was a destructive sampling study in 60 Es1$^e$ mice in which one blood sample and the corresponding brain tissue was collected at a single time point from each animal. Both male and female mice are included in the study (randomly distributed). The plot of 10 time points between 0- and 180-minutes post DMF dosing are shown in Fig 5 (DMF plasma concentrations) and Fig 6 (MMF plasma concentrations). In Fig 5, the DMF concentrations in the Control group are below the limit of detection, a finding that is consistent with the human disposition of DMF after an oral dose in which only MMF can be detected in

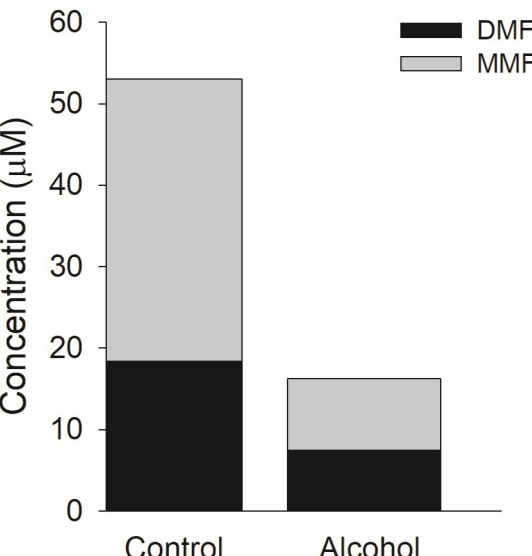

**Fig 4. Hydrolysis of DMF in CES1 with alcohol produces unknown metabolite(s).** DMF (50 μM) was incubated in human recombinant CES1 for 60 minutes and the total amount of DMF and MMF remaining at the end of the incubation was quantified. Compared to the Control, Alcohol (200 mM) resulted in a decrease in the formation of MMF. However, DMF in the incubation containing alcohol was lower than in Control indicating that hydrolysis products other than MMF were formed. Both the DMF and MMF concentrations were statistically different, with p<0.05 between the Control and Alcohol group.

plasma. In contrast, the Alcohol group produces detectable concentration of DMF with an AUC of DMF comparable to the MMF AUC (32116 versus 30344 nmole/kg*min, respectively). In Fig 6, the MMF Cmax (500 nmole/L versus 1614 nmole/L) and AUC (30344 nmole/L*min versus 46669 nmole/L*min) are lower in the Alcohol group compared to the Control group (see Table 2).

The concentrations of MMF in the brain tissue are expressed as nmole/Kg wet weight of brain tissue and parallel the plasma concentrations with significantly higher exposure of MMF in the Control group versus the Alcohol group (Fig 7). The Alcohol group had quantifiable concentrations of DMF in the brain tissue, but they were extremely low given the concentrations achieved in the plasma. In comparing the ratio of brain to plasma concentration in Table 2, MMF appeared to penetrate into brain tissue far more efficiently (AUC brain to plasma ratio = 0.60) than DMF (AUC brain to plasma ratio = 0.013).

## Discussion

The United States Food and Drug Administration prescribing information for Tecfidera™ (extended-release formulation of DMF, Biogen, Cambridge, MA) states that DMF is completely hydrolyzed to MMF prior to reaching the systemic circulation by esterases that are ubiquitous in the gastrointestinal tract, blood, and tissues. However, our studies in human recombinant CES1 and CES2 demonstrate that the hydrolysis of DMF to MMF is due to CES1 enzyme activity and CES2 plays no significant role in the conversion of DMF to its active metabolite. Corroborating this is the lack of significant DMF hydrolysis in HIM (Fig 2), which has the highest level of CES2 expression in the body [6]. The lack of activity in HIM also indicates that other esterases in the intestinal tissue do not contribute to DMF hydrolysis to MMF. The concentration-dependent inhibition of DMF hydrolysis by alcohol in human recombinant CES1 (Fig 3) and inhibition of MMF formation in Es1[e] mice dosed with alcohol (Fig 6) further

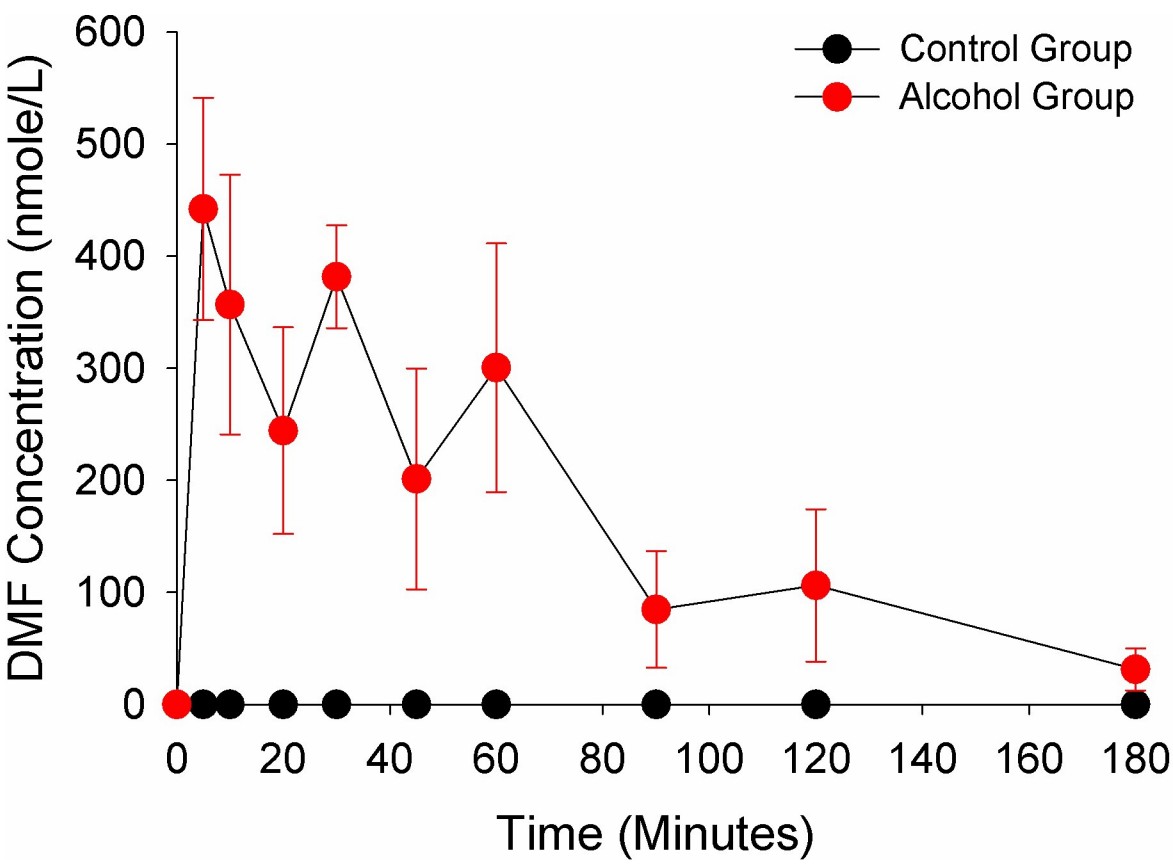

**Fig 5. DMF plasma concentrations in control and alcohol group.** Each time point represents the mean plasma concentration from three mice with the bars indicating the standard deviation. All concentration-time points were statistically different between the Control and Alcohol group (p< 0.05 with a Benjamini-Hochberg Procedure for multiple comparisons).

supports CES1-mediated hydrolysis as alcohol is a well-known and specific inhibitor of CES1 enzyme activity [24]. Thus, DMF is primarily hydrolyzed by hepatic CES1 and hydrolysis in gastrointestinal tissue or other tissues is unlikely to play any significant role in the formation of the active metabolite.

The effect of alcohol on the disposition of the active MMF metabolite in Es1$^e$ mice was extensive. After the DMF dose in the control group only MMF was quantifiable in the plasma, which is consistent with the disposition in human subjects [2]. When alcohol is administered prior to the DMF dose, the Cmax and AUC of MMF in the plasma declined 69% and 39%, and in the brain declined 50% and 20%, respectively. But the most striking change in disposition was the DMF plasma AUC which went from zero in the control group to 32116 nmole/L*min in the alcohol group. The AUC for DMF was greater than the AUC for MMF in the plasma in the mice receiving alcohol. This enhanced DMF exposure in the plasma might be expected to contribute to the therapeutic effect because DMF is a potent inducer of the Nuclear factor (ery-throid-derived 2)-like 2 (Nrf-2) pathway believed to underlie the therapeutic effect of DMF in the treatment of MS [25]. However, the exposure of DMF in brain tissue was extremely low with a ratio of brain tissue to plasma AUC of 0.013 compared to 0.60 for MMF (Table 2). Thus, the high exposure of DMF in plasma did not lead to a high exposure in brain tissue, and the low exposure level of DMF in brain (430 nmole/L*min) is unlikely to contribute signifi-cantly to the therapeutic activity of MMF. Collectively, these results predict that the most likely

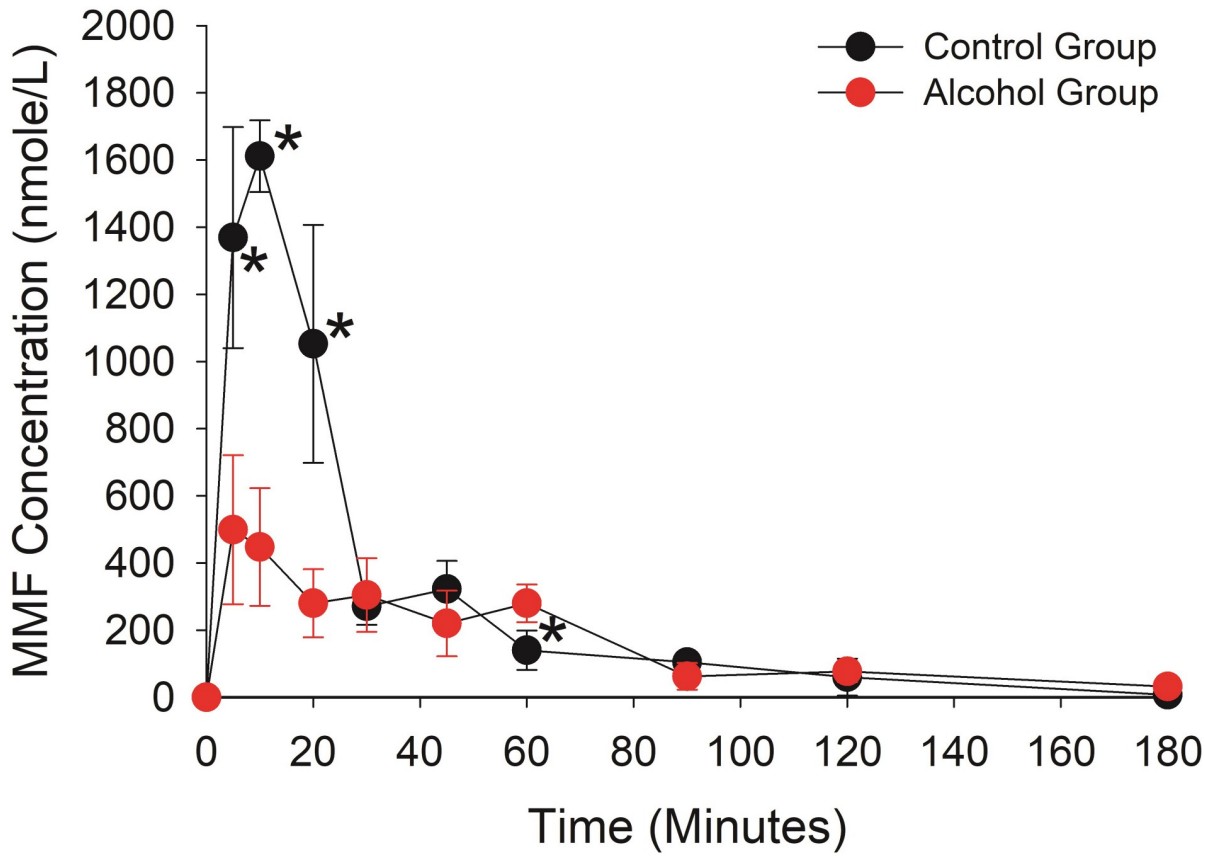

**Fig 6. MMF plasma concentrations in control and alcohol group.** Each time point represents the mean plasma concentration from three mice with the bars indicating the standard deviation. An asterisk beside the concentration-time point indicates the mean concentration difference between the Control and Alcohol group is statistically different (p< 0.05 with a Benjamini-Hochberg Procedure for multiple comparisons).

outcome of the consumption of alcohol with a DMF dose will be a decrease in exposure to the MMF active metabolite leading to reduced efficacy.

Moderate alcohol consumption is associated with a lower risk of developing MS [26, 27] and a decrease in the risk of developing severe disability in people who have MS [28]. The association

**Table 2. The Cmax and AUC of DMF and MMF in Es1^e mouse study.** The mice received water 10 minutes prior to a 100 mg/kg DMF dose (Control group) and 3 g/kg alcohol (Alcohol group) 10 minutes prior to a 100 mg/kg DMF dose. All doses were given by oral gavage. ND = not detected.

|  | Control | | Alcohol | |
|---|---|---|---|---|
| Plasma | DMF | MMF | DMF | MMF |
| Cmax (nmole/L) | ND | 1614 | 444 | 500 |
| AUC (nmole/L*min) | ND | 46669 | 32116 | 30344 |
| Brain |  |  |  |  |
| Cmax (nmole/Kg) | ND | 354 | 17 | 177 |
| AUC* (nmole/Kg*min) | ND | 22589 | 430 | 18139 |
| Ratio (Brain/Plasma) |  |  |  |  |
| Cmax | - | 0.22 | 0.038 | 0.35 |
| AUC | - | 0.48 | 0.013 | 0.60 |

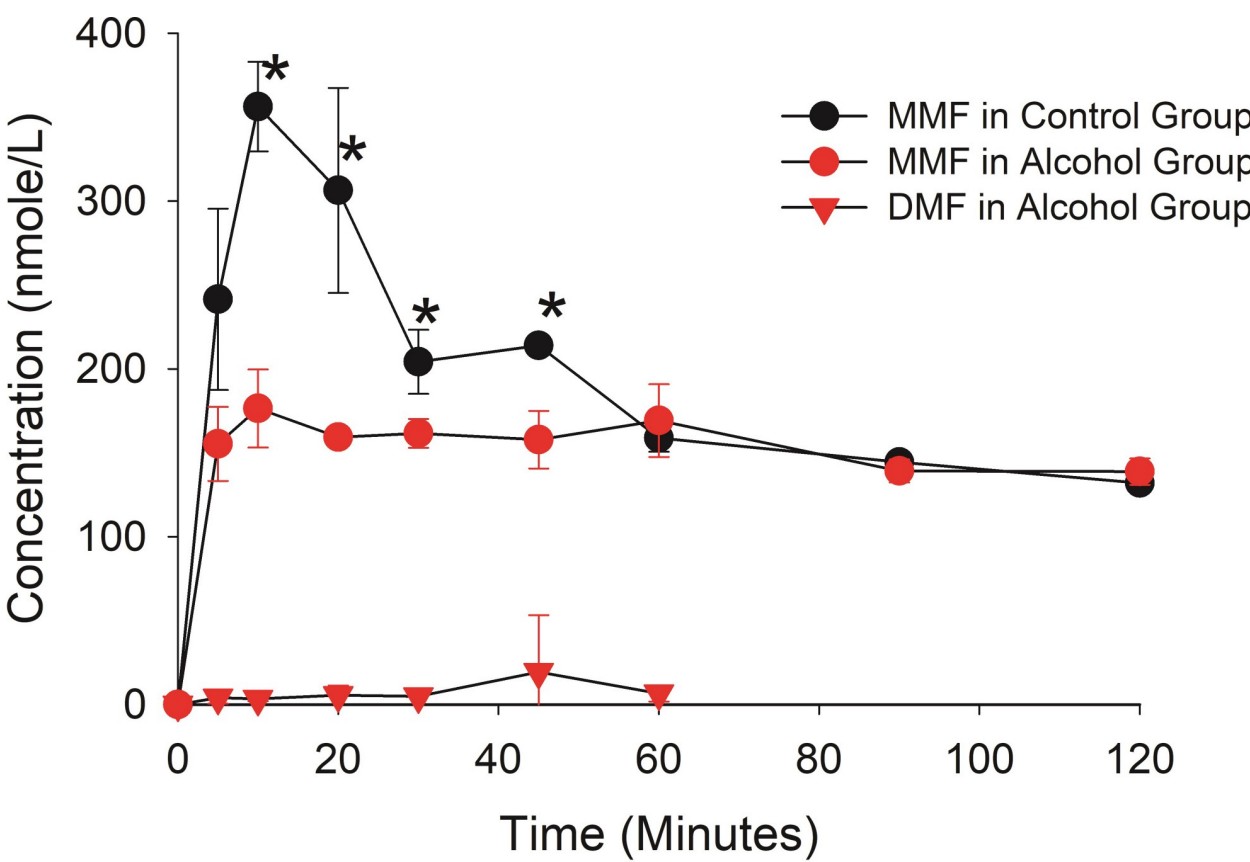

**Fig 7. The brain tissue concentrations of DMF and MMF.** The brain tissue concentrations are expressed as the nmole of DMF and MMF per Kg of brain tissue (mean ± standard deviation). An asterisk beside the concentration-time point indicates the mean concentration difference between the Control and Alcohol group is statistically different ($p < 0.05$ with a Benjamini-Hochberg Procedure for multiple comparisons).

of moderate alcohol consumption with beneficial effects on the progression of disability in MS has been questioned, but it is generally accepted that moderate alcohol consumption is not harmful in people with MS [28]. Alcohol use is common in this population with more than 60% of people with MS consuming at least low levels of alcohol (15 g/week) and over 38% with moderate alcohol consumption equivalent to up to 30 g/day in women and up to 45 g/day in men [28]. Thus, the interaction of DMF with alcohol is expected to be a common occurrence in people with RRMS taking this medication twice daily. Furthermore, a common adverse effect associated with DMF is flushing, which is reduced by consuming DMF with a meal. Thus, taking the second daily dose with dinner would be encouraged to reduce this adverse effect, and this would coincide with the daily meal most likely to include alcohol consumption.

The other aspect of the interaction of alcohol with DMF is the formation of novel metabolites formed by transesterification. The in vitro incubation of DMF and alcohol in CES1 (Fig 4) showed extensive hydrolysis of DMF with only a small amount of MMF formed suggesting that other hydrolysis products not measured by the assay were being formed. However, in the Es1[e] mouse study no transesterified metabolites were identified including EMF or DEF. It is possible that transesterified metabolites are formed but rapidly hydrolyzed leading to undetectable levels in plasma. Additionally, if EME is hydrolyzed to mono-ethyl fumarate it would not be detected by our assay. Further study is needed to elucidate the transesterification pathway of DMF.

## Conclusion

This study of the interaction between alcohol and DMF in a mouse model identifies a drastic change in the disposition of DMF and its active MMF metabolite resulting in a significant decrease in MMF exposure in the central nervous system. A decrease in MMF exposure in the central nervous system could potentially reduce the effectiveness of DMF in reducing the frequency of relapses in patients with RRMS. It is recommended based on this study that alcohol consumption be avoided in close temporal proximity to dosing with DMF to avoid this interaction. The weakness in this recommendation is that it is based on preclinical data that has not been validated by a study in human subjects. However, in our previous translational study of the interaction between alcohol and the CES1-substrate drug oseltamivir, in vitro studies in human recombinant CES1 accurately predicted the interaction in human subjects [11]. This previous work provides support that the effects of alcohol on MMF exposure demonstrated in this study are likely to occur in humans and could potentially reduce the effectiveness of DMF in the treatment of RRMS.

## Supporting information

**S1 Fig. The formation of MMF in CES1, CES2, and HIM.**
(PDF)

**S2 Fig. The concentration-dependent Inhibition of CES1 catalyzed hydrolysis by alcohol.**
(PDF)

**S3 Fig. Hydrolysis of DMF in CES1 with alcohol produces unknown metabolite(s).**
(PDF)

**S4 Fig. DMF plasma concentrations in control and alcohol group.**
(PDF)

**S5 Fig. MMF plasma concentrations in control and alcohol group.**
(PDF)

**S6 Fig. The brain tissue concentrations of DMF and MMF.**
(PDF)

## Author Contributions

**Conceptualization:** Robert B. Parker, S. Casey Laizure.

**Data curation:** Bing Yang, Robert B. Parker, Bernd Meibohm, Zaid H. Temrikar, Ashish Srivastava, S. Casey Laizure.

**Formal analysis:** Robert B. Parker, S. Casey Laizure.

**Funding acquisition:** Robert B. Parker, S. Casey Laizure.

**Investigation:** Bing Yang, Robert B. Parker, Zaid H. Temrikar, Ashish Srivastava, S. Casey Laizure.

**Methodology:** Bing Yang, Robert B. Parker, Bernd Meibohm, Zaid H. Temrikar, S. Casey Laizure.

**Project administration:** Bing Yang, S. Casey Laizure.

**Writing – original draft:** Bing Yang, Robert B. Parker, S. Casey Laizure.

**Writing – review & editing:** Bing Yang, Robert B. Parker, Bernd Meibohm, Zaid H. Temrikar, Ashish Srivastava, S. Casey Laizure.

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
