## [Decision Letter · Decision Letter 0]

9 Sep 2022

PONE-D-22-21279Alcohol Consumption may Decrease the Effectiveness of Dimethyl Fumarate for Preventing Relapses in Patients with Multiple SclerosisPLOS ONE

Dear Dr. Laizure,

Thank you for submitting your manuscript to PLOS ONE. After careful consideration, we feel that it has merit but does not fully meet PLOS ONE’s publication criteria as it currently stands. Therefore, we invite you to submit a revised version of the manuscript that addresses the points raised during the review process.

ACADEMIC EDITOR:The title uses the word patients, but no human studies have  been performed. It needs to be changed. Also include the previous experiments with dimethyl fumarate in the Introduction section. Statistical analysis needs to be re-validated. 

We look forward to receiving your revised manuscript.

Kind regards,

Kaisar Raza

Academic Editor

PLOS ONE

Journal Requirements:

"These studies were funded by the National Institute of Neurological Disorders and Stroke R03NS116229 and NIH S10 grant 1S10OD016226-01A1; Es1e mice were kindly provided by Dr. Phillip Potter, St. Jude Children’s Research Hospital, Memphis, TN"

"These studies were funded by the National Institute of Neurological Disorders and Stroke grant R03NS116229 (PI: S. Casey Laizure) and NIH S10 grant 1S10OD016226-01A1 (PI: B. Meibohm). The mice used in this study were kindly provided by Dr. Phillip Potter, St. Jude Children’s Research Hospital, Memphis, TN. The funders had no role in study design, data collection and analysis, decision to publish, or preparation of the manuscript. "

Reviewers' comments:

Reviewer's Responses to Questions

**Comments to the Author**

1. Is the manuscript technically sound, and do the data support the conclusions?

Reviewer #1: Yes

Reviewer #2: Yes

2. Has the statistical analysis been performed appropriately and rigorously? 

Reviewer #1: Yes

Reviewer #2: Yes

3. Have the authors made all data underlying the findings in their manuscript fully available?

Reviewer #1: Yes

Reviewer #2: Yes

4. Is the manuscript presented in an intelligible fashion and written in standard English?

Reviewer #1: Yes

Reviewer #2: Yes

5. Review Comments to the Author

Reviewer #1: Title of manuscript “Alcohol Consumption may Decrease the Effectiveness of Dimethyl Fumarate for Preventing Relapses in Patients with Multiple Sclerosis” is a good draft.

Comments:

1. There are large number of articles already in literature about “Dimethyl fumarate for multiple sclerosis”

Ref : https://www.cochranelibrary.com/cdsr/doi/10.1002/14651858.CD011076.pub2/full

https://www.tandfonline.com/doi/abs/10.1517/14740338.2015.977251, What is the novelty in this study?

2. Provide the clear data, effect of dimethyl fumarate for Multiple Sclerosis.

3. Abbreviation should mention at their first time appearance in the manuscript. Like DMF.

4. Recommended to design the few more 2-3 tables for the manuscript based on the discussion in the manuscript.

5. What is the comparative effectiveness of the DMF and MMF? Provide the date in table format.

6. Elaborate the supporting information.

7. Recommended to author to cite the latest publication.

8. Figure 1 is blurr, need to draw the structures.

Reviewer #2: The manuscript entitled Alcohol Consumption may Decrease the Effectiveness of Dimethyl Fumarate for Preventing Relapses in Patients with Multiple Sclerosis is a good piece of research work. Although as a reviewer I have general suggestions and queries. which are appended below:

1. In Introduction section the background about multiple sclerosis and the action of DMF in multiple sclerosis is lacking which is important for readers to understand.

2. Line 55: For hypothesis the support and reference must be sufficiently elaborated in favour of CES1.

3. Further, ruling of transesterification needs references.

4.For In vitro incubation studies please provide the reference as well as mention why the studies are conducted in duplicate not in triplicate?

5. For In vivo studies in mice the details of model followed with protocol is desirable in methods section as well as

please mention the reference for dose of 3g/kg alcohol. also, mention the quantity of retro-orbital blood samples for better understanding of readers.

6. In results in DMF alcohol study please mention the data in text for the ease.

7. The recommendation in conclusion section regarding alcohol consumption two hours before and after if supported by the result and findings from your work would be appreciated.

6. PLOS authors have the option to publish the peer review history of their article (what does this mean?). If published, this will include your full peer review and any attached files.

Reviewer #1: No

Reviewer #2: No

---

## [Author Response · Author response to Decision Letter 0]

24 Oct 2022

PONE-D-22-21279

ACADEMIC EDITOR:

The title uses the word patients, but no human studies have been performed. It needs to be changed. Also include the previous experiments with dimethyl fumarate in the Introduction section. Statistical analysis needs to be re-validated. 

The original title, “Alcohol Consumption may Decrease the effectiveness of Dimethyl Fumarate for Preventing Relapses in Patients with Multiple Sclerosis” has been revised to, “Alcohol Inhibits the Formation of Dimethyl Fumarate’s Active Metabolite Responsible for Decreasing Relapses in the Treatment of Multiple Sclerosis”.

Unclear what previous experiments you are referring to that should be put in the introduction. All the experiments described in this paper are original work that has not been published and cannot be included in the introduction section, which would precede the description of how they were done in the Methods section. 

We have revised the manuscript to include a basic statistical analysis of the results of the incubation study in the recombinant enzyme (Figure 3), and the concentration-time data from the mouse study (Figures 5, 6, and 7). A description of the statistics has been added to the Methods section. The figures have been revised to include an asterisk by the concentration-time points that are statistically different. 

Comments to the Author

1. Is the manuscript technically sound, and do the data support the conclusions?

Reviewer #1: Yes

Reviewer #2: Yes

2. Has the statistical analysis been performed appropriately and rigorously? 

Reviewer #1: Yes

Reviewer #2: Yes

3. Have the authors made all data underlying the findings in their manuscript fully available?

Reviewer #1: Yes

Reviewer #2: Yes

4. Is the manuscript presented in an intelligible fashion and written in standard English?

Reviewer #1: Yes

Reviewer #2: Yes

5. Review Comments to the Author

Reviewer #1: Title of manuscript “Alcohol Consumption may Decrease the Effectiveness of Dimethyl Fumarate for Preventing Relapses in Patients with Multiple Sclerosis” is a good draft.

Comments:

1. There are large number of articles already in literature about “Dimethyl fumarate for multiple sclerosis”

Ref : https://www.cochranelibrary.com/cdsr/doi/10.1002/14651858.CD011076.pub2/full

https://www.tandfonline.com/doi/abs/10.1517/14740338.2015.977251, What is the novelty in this study?

The novelty of this study is that it is the first to report that alcohol inhibits the formation of the active metabolite, monomethyl fumarate from dimethyl fumarate leading to a reduction in exposure to the active metabolite and quantifiable concentrations of dimethyl fumarate in the plasma. If this interaction between alcohol and dimethyl fumarate is confirmed in a human study, it would most likely require that the interaction be included in the FDA prescribing information. 

2. Provide the clear data, effect of dimethyl fumarate for Multiple Sclerosis.

The effect of dimethyl fumarate on multiple sclerosis is elaborated in reference 20. The present study is a pharmacokinetic study, and there is no data presented on the effect of dimethyl fumarate for the treatment of multiple sclerosis. 

3. Abbreviation should mention at their first time appearance in the manuscript. Like DMF.

Yes, we agree and have revised accordingly. 

4. Recommended to design the few more 2-3 tables for the manuscript based on the discussion in the manuscript.

Unclear what the reviewer is proposing to be included in these tables. 

5. What is the comparative effectiveness of the DMF and MMF? Provide the date in table format.

This is unknown as DMF has never been clinically evaluated in the treatment of multiple sclerosis since it is impossible to achieve quantifiable levels in humans. It is theorized that DMF would have an effect on multiple sclerosis if it achieved significant plasma concentrations based on its proposed effect on the Nrf2 pathway demonstrated in in vitro studies; however, this is hypothetical. This study only demonstrates that alcohol drastically changes the disposition of DMF. How these changes other than the decrease in active metabolite alter its therapeutic activity are speculative and beyond the scope of this study. 

6. Elaborate the supporting information.

A supplemental supporting information documents has been added. 

7. Recommended to author to cite the latest publication.

The reviewer would have to be more specific for me to address this issue. 

8. Figure 1 is blurr, need to draw the structures.

I think the concatenation in forming the pdf led to a decrease in resolution. However, we will consult with editors to ensure the figure is adequate. 

Reviewer #2: The manuscript entitled Alcohol Consumption may Decrease the Effectiveness of Dimethyl Fumarate for Preventing Relapses in Patients with Multiple Sclerosis is a good piece of research work. Although as a reviewer I have general suggestions and queries. which are appended below:

1. In Introduction section the background about multiple sclerosis and the action of DMF in multiple sclerosis is lacking which is important for readers to understand.

We have added a statement in the introduction on the specific therapeutic effect of DMF in the treatment of multiple sclerosis.

2. Line 55: For hypothesis the support and reference must be sufficiently elaborated in favour of CES1.

Previous support for our hypothesis from the literature does not exist. The data presented in this manuscript in recombinant enzymes and HIM provide the support that DMF is a CES1 substrate and was used to justify conducting the study in mice. The results from the mouse study confirm that it is a CES1 substrate, which is susceptible to inhibition by alcohol, a well-known CES1 enzyme inhibitor. 

3. Further, ruling of transesterification needs references.

I am unclear as to what the reviewer means by “ruling”.

4.For In vitro incubation studies please provide the reference as well as mention why the studies are conducted in duplicate not in triplicate?

All the incubation studies were conducted in triplicate except for the initial qualitative study in CES1, CES2, and HIM (Figure 2) in which each time point represents a single incubation. We have revised the text in the manuscript to reflect this and we have also provided a supplementary data file giving the raw data used to construct the figures. It is unclear what the reviewer is asking us to reference. 

5. For In vivo studies in mice the details of model followed with protocol is desirable in methods section as well as please mention the reference for dose of 3g/kg alcohol. also, mention the quantity of retro-orbital blood samples for better understanding of readers.

We have added statements to the In Vivo Study in the Es1e Mice section in Methods referencing the Es1e mouse model, and the dose of DMF and alcohol used in the 

mouse study. The blood volume collected has been added to the text. 

6. In results in DMF alcohol study please mention the data in text for the ease.

The data in Table 2 is referenced in the text and we have added the specific data in the text on the comparison of the DMF and MMF AUC in the alcohol group from the table. 

7. The recommendation in conclusion section regarding alcohol consumption two hours before and after if supported by the result and findings from your work would be appreciated.

We have revised this section of the discussion to better reflect the data from this study by removing the specific time recommendation and simply indicating that consumption of alcohol close to DMF dosing may decrease MMF exposure. 

6. PLOS authors have the option to publish the peer review history of their article (what does this mean?). If published, this will include your full peer review and any attached files.

---

## [Decision Letter · Decision Letter 1]

10 Nov 2022

Alcohol Inhibits the Metabolism of Dimethyl Fumarate to the Active Metabolite Responsible for Decreasing Relapse Frequency in the Treatment of Multiple Sclerosis

PONE-D-22-21279R1

Dear Dr. Laizure,

We’re pleased to inform you that your manuscript has been judged scientifically suitable for publication and will be formally accepted for publication once it meets all outstanding technical requirements.

Kind regards,

Kaisar Raza

Academic Editor

PLOS ONE

Additional Editor Comments (optional):

Reviewers' comments:

Reviewer's Responses to Questions

**Comments to the Author**

1. If the authors have adequately addressed your comments raised in a previous round of review and you feel that this manuscript is now acceptable for publication, you may indicate that here to bypass the “Comments to the Author” section, enter your conflict of interest statement in the “Confidential to Editor” section, and submit your "Accept" recommendation.

Reviewer #1: All comments have been addressed

Reviewer #2: All comments have been addressed

2. Is the manuscript technically sound, and do the data support the conclusions?

Reviewer #1: Yes

Reviewer #2: Yes

3. Has the statistical analysis been performed appropriately and rigorously? 

Reviewer #1: Yes

Reviewer #2: Yes

4. Have the authors made all data underlying the findings in their manuscript fully available?

Reviewer #1: Yes

Reviewer #2: Yes

5. Is the manuscript presented in an intelligible fashion and written in standard English?

Reviewer #1: Yes

Reviewer #2: Yes

6. Review Comments to the Author

Reviewer #1: The Authors addressed all comments to the reviewer's so the manuscript publishable in present manner.

Reviewer #2: The authors have successfully provided the rebuttal. The corrections are incorporated satisfactorily. The findings are supported by literature.

7. PLOS authors have the option to publish the peer review history of their article (what does this mean?). If published, this will include your full peer review and any attached files.

Reviewer #1: No

Reviewer #2: No

---

## [Editor Report · Acceptance letter]

15 Nov 2022

PONE-D-22-21279R1 

Alcohol inhibits the metabolism of dimethyl fumarate to the active metabolite responsible for decreasing relapse frequency in the treatment of multiple sclerosis 

Dear Dr. Laizure:

I'm pleased to inform you that your manuscript has been deemed suitable for publication in PLOS ONE. Congratulations! Your manuscript is now with our production department. 

Kind regards, 

on behalf of

Dr. Kaisar Raza 

Academic Editor

PLOS ONE